# The benefits of higher LMR for early threatened abortion: A retrospective cohort study

Qiu-Ting Feng[1], Chi Chen[2], Qing-Ying Yu[1], Si-Yun Chen[1], Xian Huang[1], Yan-Lan Zhong[1], Song-Ping Luo[1,3], Jie Gao[1,3]*

1 Guangzhou Univ Chinese Med, Guangzhou, China, 2 Guiyang College of Traditional Chinese Medicine, Guiyang, China, 3 The First Affiliated Hospital of Guangzhou University of Chinese Medicine, Guangzhou, China

* gjfkts@qq.com

## Abstract

### Problem

To investigate the relation of inflammation-related parameters and pregnancy outcome in women with the early threatened abortion.

### Method of study

630 women with early threatened abortion were divided into two groups based on the pregnancy outcome. All of them had the blood routine examination before treating. The differences between two groups were analyzed by the Chi-squared test, Student T test, Mann-Whitney U test, Binary Logistic Regression, Marginal Structural Model and Threshold effect analysis.

### Results

We found that there is no significant difference in the pregnancy outcome for NLR (OR:0.92, CI95%:0.72, 1.17) and PLR (OR:1.00, CI%:0.99, 1.01). However, a difference had a statistical significance in the pregnancy outcome when LMR less than 2.19 (OR:0.39, CI95%:0.19,0.82).

### Conclusions

This study suggested that higher LMR was related to the lower risk of miscarriage in the women with early threatened abortion in a way.

## Introduction

Threatened abortion, accounting for about 30–40% in the pregnancies, is diagnosed with the clinical symptoms of vaginal spotting or bleeding within 20 weeks of gestation [1].

**Data Availability Statement:** All relevant data are within the paper and its Supporting Information file.

**Funding:** National Natural Science Foundation of China (No.81774358); Fok Ying-Tong Education Foundation of China (No.151042); Pearl River Scholars, Department of Education, Guangdong Province (Guangdong teacher's letter[2017] number79); Scientific Research Team Training Project of GZUCM(2019KYTD202); Guangzhou Univ Chinese Med Planning[2019]5 (XK2019016)

**Competing interests:** The authors have declared that no competing interests exist.

Furthermore, approximately half of women with threatened abortion will suffer from a subsequent miscarriage, especially with higher incidence in the first trimester [2,3,4]. In China, the number of gravidas is largely increasing due to the *Two Child Policy*, it means the more women with threatened abortion, the more women who will suffer from pregnancy loss [5]. The negative effects of adverse pregnancy outcome cannot be ignored for individual and even society [6]. Therefore, the more advanced arithmetic skills are required to stratify the risk of miscarriage following early threatened abortion.

Up to date, the discovery of miscarriage-associated risk factors is still on the way. Current known risk factors include certain infections [7], endocrine disturbance [8], genital malformation [9], chromosome aberrations [10] and immune system disorder [11]. Recently, the relationship between inflammation and miscarriage has attracted more attention. The evidence obtained from experimental studies has suggested that inflammation is involved in the whole pregnancy evolution [12,13,14]. In addition, the clinical studies have found that inflammation-related parameters are strongly correlated with the pregnancy complications [15,16]. However, the evidence of the link between inflammation and miscarriage in women with early threatened abortion is limited. Given that the similar maternal-fetal pathological changes between threatened abortion and miscarriage [17,18], we speculated that inflammation might be associated with pregnancy outcome in the early threatened abortion.

Neutrophil-lymphocyte ratio (NLR), platelet-lymphocyte ratio (PLR) and lymphocyte-monocyte ratio (LMR) are a series of new parameters derived from complete blood counting reflecting inflammatory status. Because of its availability and extensive use, accumulating studies have been reported that these inflammation-related parameters are pretty relative with the prognosis and genesis of many diseases [19,20]. Therefore, we sought to evaluate the associations of aforementioned indexes and the early pregnancy outcome in women with early threatened abortion.

## Materials and methods

### Study population

We initially collected a total of 765 women who were diagnosed as early threatened abortion between June 2010 and December 2018 at the First Affiliated Hospital of Guangzhou University of Chinese Medicine. All of them had received progesterone treatment according to the guidelines of European Progestin Club (EPC). Early threatened abortion is the clinical term for vaginal spotting or bleeding within 12 weeks of gestation[4]. Due to the small sample size and the missing information, we strictly screened the objects according the following exclusion criteria: (1) age is either less than 18 or greater than 40. (2) be exposed to chronic alcohol or nicotine. (3) suffer from chronic systemic diseases or acute infectious illness. (4) reproductive tract malformation. (5) multiple gestation. (6) loss of follow-up. After the screening, 630 women were selected for final analysis (Fig 1). All participants were followed up in 12 weeks of gestation by either a subsequent ultrasound scan or a telephone interview.

Because the detailed information is anonymously collected from hospital electronic medical record system, the written informed consents were not required. The study was approved by the ethics committee of the First Affiliated Hospital of Guangzhou University of Chinese Medicine.

### Variables

We obtained NLR, LMR and PLR at baseline, which were recorded as continuous variables. The detailed process was described as follows: (1) we pumped patient's fresh peripheral venous blood into EDTA-anticongulation tube on the second day morning of the hospitalization. (2)

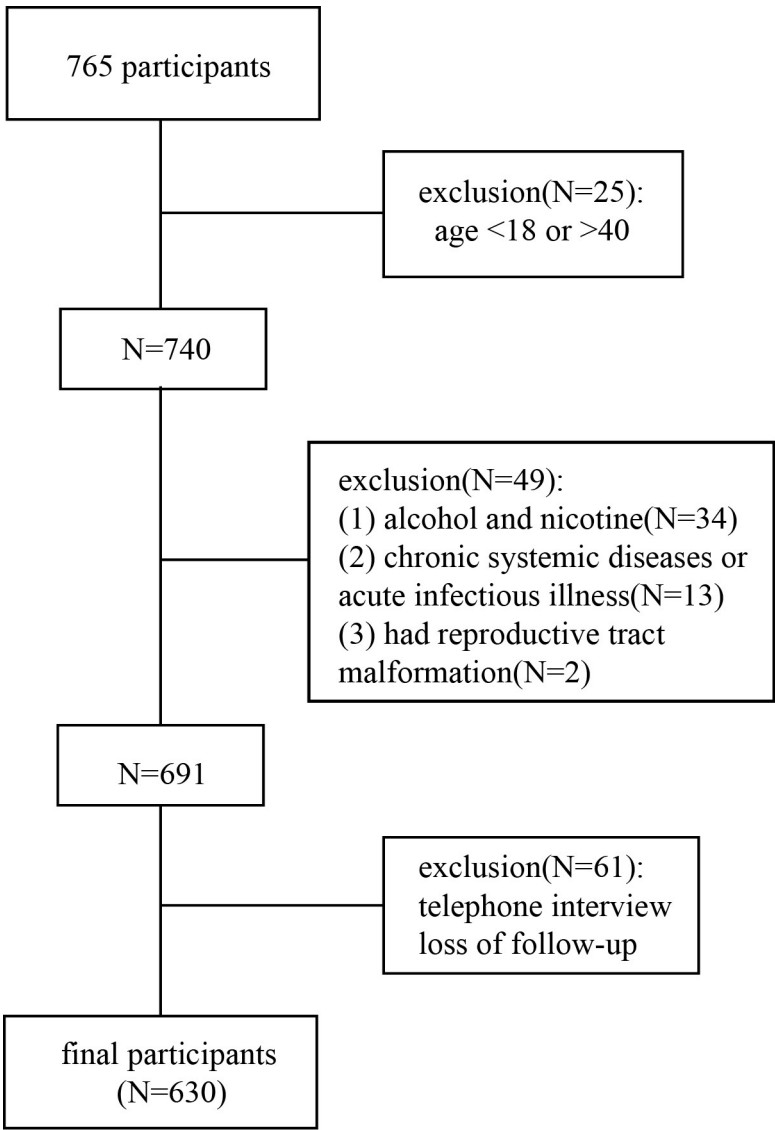

**Fig 1. Flowchart of screening participants.**

samples were sent to central lab and tested by an automatic blood cell analysis instrument (UniCel DxH800).

According to the published guidelines and researches, we decided to obtain this outcome variable: the early pregnancy outcome (dichotomous variable). The detailed definition was described as follows: (1) the survival of embryo within 12 weeks of gestation, namely successful pregnancy. (2) conversely, miscarriage occurs within 12 weeks of gestation, namely failed pregnancy.

The covariates in this study can be classified as follows: (1) demographic data; (2) basing on our clinical experiences, variables that might affect exposure and outcome variables, were reported by previous literature. Therefore, the following variables (obtained at baseline) were used to construct the fully-adjusted model: (1) continuous variables: age(years), body mass index(BMI, kg/m$^2$), gestational week(weeks); hemoglobin(HGB, g/L); β human chorionic gonadotropin (β-HCG, nmol/L); estradiol(E$_2$, pmol/L); progesterone(P, nmol/L); (2)

categorical variables: marital status(yes/no); childbearing history(yes/no); spontaneous abortion history(yes/no); artificial abortion history(yes/no).

## Statistical analysis

In this study, our presentation of continuous variables was primarily based on whether they were normally distributed. If it was a normal distribution, we present the continuous variable as mean (SD), and vice versa as the medium (min, max). Categorical variables were expressed as a percentage. We used χ2 (categorical variables), Student T test (normal distribution), or Mann-Whitney U test (skewed distribution) to test for differences between two groups. The process of entire data analysis mainly can be divided into two steps. Step 1: Univariate and multivariate binary logistic regression were employed. We constructed two models: model 1, no covariates were adjusted; model 2, adjusted for sociodemographic data presented in Table 1. Step 2: To address for non linearity of exposure and outcome variables, a generalized additive model and smooth curve fitting (penalized spline method) were conducted. If

**Table 1. Baseline characteristics of all participants.**

| Characteristics | All participants |
|---|---|
| Age, mean (SD), years | 30.41±4.73 |
| Gestational week, medium (IQR), weeks | 6.00 (5.00–7.00) |
| HGB, medium (IQR), g/L | 122.00 (115.00–130.00) |
| β-HCG, medium (IQR), nmol/L | 15455.00 (2596.00–60113.50) |
| P, medium (IQR), nmol/L | 81.65 (58.23–114.10) |
| E2, medium (IQR), pmol/L | 1818.50 (835.80–3184.50) |
| BMI, mean (SD), kg/m$^2$ | 21.03 (2.84) |
| WBC, medium (IQR), $10^9$/L | 7.98 (6.63–9.50) |
| Eosinophils, medium (IQR), $10^9$/L | 0.11 (0.07–0.17) |
| Neutrophil, medium (IQR), $10^9$/L | 5.25 (4.04–6.49) |
| Lymphocyte, medium (IQR), $10^9$/L | 1.98 (1.66–2.39) |
| Monocyte, medium (IQR), $10^9$/L | 0.47 (0.38–0.55) |
| Platelet, medium (IQR), $10^9$/L | 238.00 (206.00–270.00) |
| NLR, medium (IQR) | 2.54 (1.99–3.30) |
| PLR, medium (IQR) | 119.37 (97.27–142.07) |
| LMR, medium (IQR) | 4.32 (3.56–5.41) |
| Marital status | |
| unmarried | 2.53% |
| married | 97.47% |
| Childbearing history | |
| no | 69.32% |
| yes | 30.68% |
| Artificial abortion history | |
| no | 70.34% |
| yes | 29.66% |
| outcome | |
| Successful pregnancy | 75.40% |
| Failed pregnancy | 24.6% |

SD, standard deviation; IQR, interquartile range; HGB, hemoglobin; β-HCG, β human chorionic gonadotropin; P, progesterone; E$_2$, estradiol; BMI, body mass index; WBC, white blood cells; NLR, neutrophil-lymphocyte ratio; PLR, platelet-lymphocyte ratio; LMR, lymphocyte-monocyte ratio.

nonlinearity was detected, we first calculated the inflection point using recursive algorithm, and then constructed a two-piecewise logistic regression on both sides of the inflection point. We determined the best fit model basing on the *P* values for log likelihood ratio test. We conducted a sensitivity analysis using marginal structural model because serum hormone and treatment are time-varied. All the analyses were performed with the statistical software packages R (http://www.R-project.org, The R Foundation) and EmpowerStats (http://www.empowerstats.com, X&Y Solutions, Inc, Boston, MA). *P*<0.05 (two-sided) was considered to be statistically significant.

## Results

### Population characteristics

After screening, a total of 630 selected participants(aged 30.41±4.73) with early threatened abortion were analyzed. 155 women (24.60%) lost pregnancy during the first trimester (medium:6.00; IQR:5.00–7.00). The exposed variables were described as following: NLR, medium (IQR): 2.54 (1.99–3.30); LMR, medium (IQR): 4.32 (3.56–5.41); PLR, medium (IQR): 119.37 (97.27–142.07). The detailed baseline characteristics of all participants were shown in the Table 1.

### Relationships of three inflammation-related parameters and early pregnancy outcome

We analyzed the associations of three continuous exposure variables and outcome variable by using non/fully-adjusted model and MSM. These models showed no significant correlations of pregnancy outcome and parameters including NLR (0.89 (0.77, 1.03), 0.92 (0.72, 1.17), 0.74 (0.45, 1.21)), LMR (0.92 (0.75, 1.12), 0.95 (0.61, 1.47), 1.04 (0.89, 1.23)) and PLR (1.00 (1.00,

**Table 2. Results of unvariate and multivariate analysis using binary logistic regression and marginal structural model.**

| Exposure | Non-adjusted model (OR, 95%CI, *P* value) | Fully-adjusted model (OR, 95%CI, *P* value) | Marginal Structural Model (OR, 95%CI, *P* value) |
|---|---|---|---|
| NLR | 0.89 (0.77, 1.03) 0.1174 | 0.92 (0.72, 1.17) 0.4883 | 0.74 (0.45, 1.21) 0.231 |
| Low | Ref | Ref | |
| Middle | 0.56 (0.36, 0.86) 0.0091 | 0.82 (0.45, 1.52) 0.5383 | |
| High | 0.56 (0.36, 0.87) 0.0103 | 0.63 (0.31, 1.26) 0.1879 | |
| *P* for trend of NLR | 0.0077 | 0.1882 | |
| LMR | 0.92 (0.75, 1.12) 0.4115 | 0.95 (0.61, 1.47) 0.8075 | 1.04 (0.89, 1.23) 0.619 |
| Low | Ref | Ref | |
| Middle | 0.52 (0.33, 0.82) 0.0047 | 0.47 (0.24, 0.93) 0.0310 | |
| High | 0.58 (0.37, 0.90) 0.0151 | 0.44 (0.17, 1.11) 0.0830 | |
| *P* for trend of LMR | 0.0176 | 0.0620 | |
| PLR | 1.00 (1.00, 1.01) 0.2859 | 1.00 (0.99, 1.01) 0.9303 | 1.00 (0.98, 1.01) 0.884 |
| Low | Ref | Ref | |
| Middle | 0.99 (0.63, 1.55) 0.9745 | 1.02 (0.55, 1.92) 0.9391 | |
| High | 1.11 (0.71, 1.72) 0.6522 | 0.86 (0.44, 1.67) 0.6511 | |
| *P* for trend of PLR | 0.6518 | 0.6615 | |

NLR, neutrophil-lymphocyte ratio; PLR, platelet-lymphocyte ratio; LMR, lymphocyte-monocyte ratio.

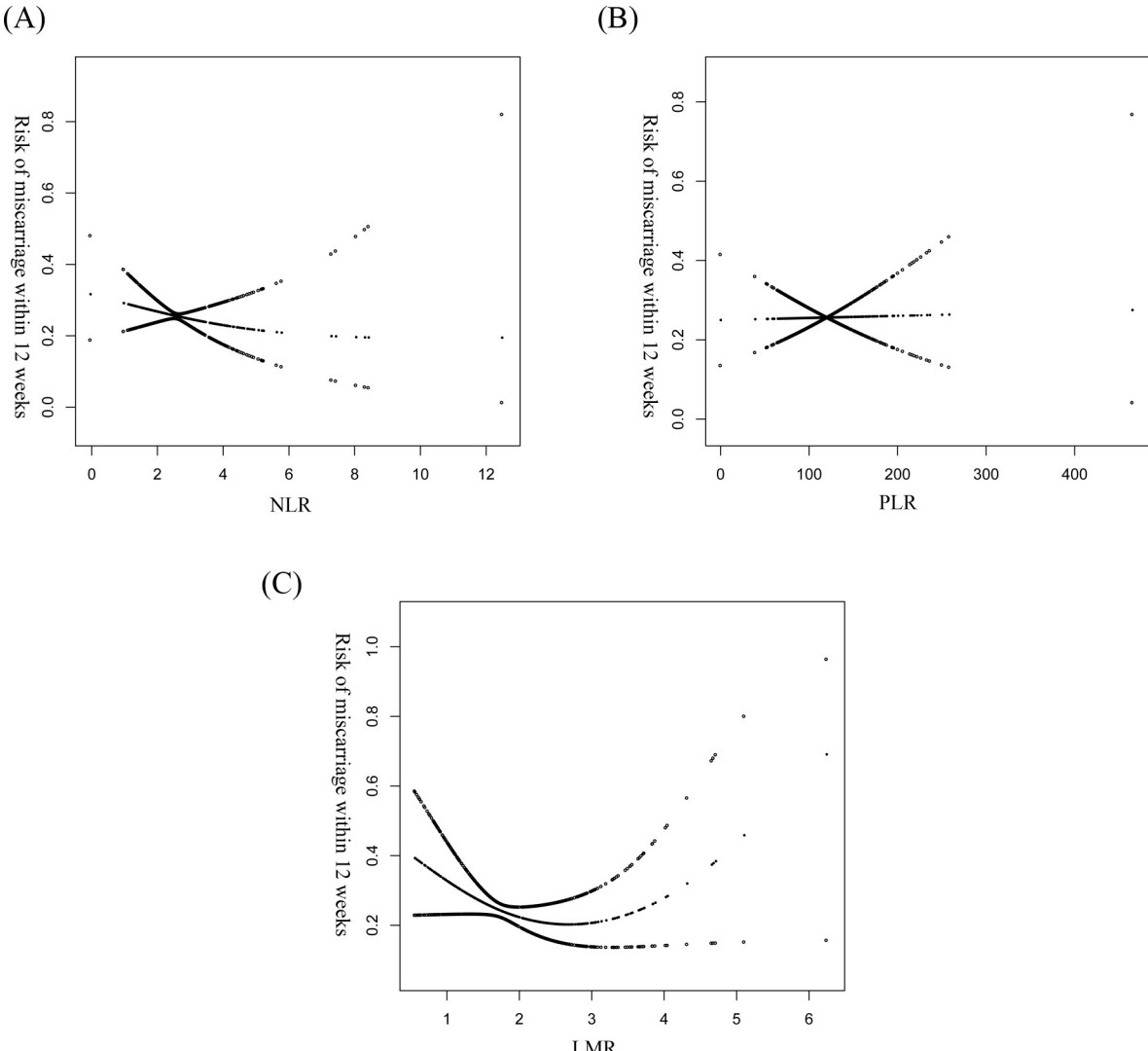

**Fig 2. The relationship between NLR/LMR/PLR and the risk of miscarriage in early women with early threatened abortion.** (A)The associatioin of NLR and the early pregnancy outcome is analyzed by GAM model and smooth fitting. (B)The associatioin of PLR and the early pregnancy outcome is analyzed by GAM model and smooth fitting. (C)The associatioin of LMR and the early pregnancy outcome is analyzed by GAM model and smooth fitting.

1.01), 1.00 (0.99, 1.01), 1.00 (0.98, 1.01)). Though there was a little difference in the ORs and 95%CIs of outcome for three inflammation-related parameters, the results were stable in all models. As tertiles, the counterpart for NLR, LMR and PLR were respectively 1.00(reference), 0.82 (0.45, 1.52), 0.63 (0.31, 1.26); 1.00(reference), 0.47 (0.24, 0.93), 0.44 (0.17, 1.11); 1.00(reference), 1.02 (0.55, 1.92, 0.86 (0.44, 1.67) in low, middle and high tertiles, which presenting a non-equidistant relation for three parameters in model 2. The detailed data were presented in the Table 2.

## Analysis of non-linear relationship

We continued to explore the potential non-linear relationship of three parameters and early pregnancy outcome using GAM model and smooth fitting(penalty curve method), as presented in the Fig 2. Non-linear relationship of LMR and early pregnancy outcome was

**Table 3. Analysis of non-linear relationship between three parameters and outcome.**

| Exposure | NLR | LMR | PLR |
|---|---|---|---|
| Fitting model using standard binary l,ogistic regression model | 0.92 (0.72, 1.17) 0.4883 | 0.95 (0.61, 1.47) 0.8075 | 1.00 (0.99, 1.01) 0.9303 |
| Fitting model using two-piecewise regression model | | | |
| Inflection point | 3.93 | 2.19 | 72.35 |
| < inflection point | 0.73 (0.51, 1.05) 0.0925 | 0.39 (0.19, 0.82) 0.0128 | 1.08 (0.92, 1.26) 0.3324 |
| ≥ inflection point | 1.22 (0.84, 1.77) 0.2872 | 1.75 (1.00, 3.06) 0.0518 | 1.00 (0.99, 1.01) 0.7360 |
| *P* for log likelyhood ratio test | 0.109 | 0.004 | 0.157 |

NLR, neutrophil-lymphocyte ratio; PLR, platelet-lymphocyte ratio; LMR, lymphocyte-monocyte ratio.

detected. We calculated an inflection point for LMR (2.19) by two-piecewise linear regression model and recursive algorithm. When LMR< 2.19, significant negative association was observed for early pregnancy outcome (OR: 0.73, 95%CI: 0.51, 1.05, *P*: 0.0128). When LMR≧2.19, insignificant positive association was observed (OR: 1.22, 95%CI: 0.84, 1.77, *P*: 0.0518). There was a threshold effect between LMR and early pregnancy outcome, which presented in the Table 3.

## Discussion

In this study, no linear relationship between NLR/PLR/LMR and early pregnancy outcome was confirmed by fully-adjusted model and MSM. However, as tertiles, we found that a significant negative association was only detected in the middle tertile of LMR (OR: 0.47, 95%CI: 0.24, 0.93, *P*: 0.0310). Furthermore, non-linearity discovered by GAM, smooth curve fitting and two-piecewise model suggested that threshold effect can be observed in LMR, but not NLR and PLR after full adjustment. At the range of less than 2.19, increasing of one unit of LMR was significantly associated with the decreasing risk of miscarriage for 61%. There was no significant association between LMR and early pregnancy outcome at the range of more than 2.19.

The blood routine examination is a easily measurable and available method to reflect the general inflammatory status [21]. Their deuterogenic parameters, such as NLR, LMR and PLR, were considered as the inflammation-related indicators with respect to predict many diseases, including pregnancy complications [22,23]. Several studies, sample size ranging from 107–814, suggested that maternal NLR/PLR was positive associated with severe pre-eclampsia/gestational diabetes mellitus/acute appendicitis during pregnancy [24,25,26]. However, some considered that NLR/PLR has no significant association with gestational diabetes mellitus and HELLP syndrome [27,28]. In addition, the maternal NLR/PLR of the six gestational week was suggested as a predictor of miscarriage, but the results were inconsistent with our study [29]. Comparing with them, our results are reliable enough because they are based on a larger sample size and added covariates. But there is an increasing focus on the relevance of inflammatory parameters and miscarriage, we need a further study to verify the associations of NLR/PLR and miscarriage.

Furthermore, we found that the higher LMR was related to the lower risk of miscarriage in women with early threatened abortion at the range of less than 2.19, which was not reported yet in the study of pregnancy-related complications. It was the first study to evaluate the relationship of LMR and miscarriage in women with early threatened abortion. Nevertheless, the

bulk of literature have described that the higher LMR was a negative factor for the prognosis of diseases, such as gastric cancer [30] and early-stage Hodgkin lymphoma [31]. But a meta analysis study has suggested that the low LMR was associated with the poor outcome for patients with esophageal squamous cell carcinoma [32]. The results of the association of maternal LMR and pregnancy outcome is not exactly consistent with some studies. As an inflammatory factor, LMR might play an important role during the gestational period. Osmanağaoğlu, M.A [33] have described that the autophagy level of peripheral blood mononuclear cells is higher in nulliparous women with miscarriage, comparing with the normal pregnancy. In addition, accumulating researches have proved that immune system disorder and abnormal inflammation leads to miscarriagel [34,35]. Songcun Wang [36] etc. have reported that *Tim-3*, a regulator for inflammatory response, promotes the production of Th-2 type cytokines during the pregnancy, which might be a potential drug target for the treatment of miscarriage. In summary, both inflammation and immunology play an important role in maintenance of pregnancy.

Clinically, anti-inflammation is a way to prevent miscarriage. Because of its availability and low price, LMR can be widely used to evaluate the risk of miscarriage in women with early threatened abortion. However, there was some limitations in this study. For the retrospective cohort study, not all the confounders were included. And larger patient population is needed to verify the results of our study.

For the present study, we concluded that the higher maternal LMR is associated with the lower risk of miscarriage in women with early threatened abortion at the range of less than 2.19.

## Supporting information

**S1 Data.**
(CSV)

## Acknowledgments

Thanks for Guangzhou Univ Chinese Med, Guangzhou, China; the First Affiliated Hospital of Guangzhou University of Chinese Medicine, Guangzhou, China; Lingnan Medical Research Center of Guangzhou University of Chinese Medicine.

## Author Contributions

**Data curation:** Qing-Ying Yu, Si-Yun Chen, Xian Huang, Yan-Lan Zhong.

**Funding acquisition:** Jie Gao.

**Methodology:** Chi Chen.

**Project administration:** Song-Ping Luo, Jie Gao.

**Writing – original draft:** Qiu-Ting Feng.

**Writing – review & editing:** Qiu-Ting Feng.

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
