## [Decision Letter · Decision Letter 0]

27 Jan 2020

PONE-D-19-27160

the Benefits of Higher LMR for Early Threatened Abortion: a Retrospective Cohort Study

PLOS ONE

Dear Gao,

Thank you for submitting your manuscript to PLOS ONE. After careful consideration, we feel that it has merit but does not fully meet PLOS ONE’s publication criteria as it currently stands. There were major comments that arose during the review of your manuscript. Therefore, we invite you to submit a revised version of the manuscript that addresses ALL of the points raised during the review process.

We would appreciate receiving your revised manuscript by Mar 12 2020 11:59PM. To enhance the reproducibility of your results, we recommend that if applicable you deposit your laboratory protocols in protocols.io, where a protocol can be assigned its own identifier (DOI) such that it can be cited independently in the future. For instructions see: http://journals.plos.org/plosone/s/submission-guidelines#loc-laboratory-protocols

We look forward to receiving your revised manuscript.

Kind regards,

Frank T. Spradley

Academic Editor

PLOS ONE

Reviewers' comments:

Reviewer's Responses to Questions

**Comments to the Author**

1. Is the manuscript technically sound, and do the data support the conclusions?

Reviewer #1: Yes

2. Has the statistical analysis been performed appropriately and rigorously? 

Reviewer #1: Yes

3. Have the authors made all data underlying the findings in their manuscript fully available?

Reviewer #1: Yes

4. Is the manuscript presented in an intelligible fashion and written in standard English?

Reviewer #1: Yes

5. Review Comments to the Author

Reviewer #1: Comments to PONE-D-19-27160 entitled the Benefits of Higher LMR for Early Threatened Abortion: a Retrospective Cohort Study

The authors retrospectively tested the inflammasome obtained from the routine whole blood cells test in the prediction of outcome of women with threatened abortion, and a total of 630 women were enrolled into the current study.

Among these, 155 women were finally complicated with early pregnancy loss compared to 475 women continued pregnancy. Some comments are shown below.

1. Although the authors clarified the meaning of abbreviation, some are not a good candidate for the abbreviation. For example, TA might not be well accepted in the clinical use. In the introduction, TA as an initial word might not be appropriate.

2. The use of simple hematological parameters to predict the various kinds of the diseases has been evaluated in the many studies. Nearly almost studies favored their clinical value. It is hard for me to believe it, although the aforementioned data have supported it. Therefore, some data had better be provided in the Table 1. Absolute number of white cells, absolute number of each subpopulation of white cells, such as neutrophil, lymphocyte, monocyte, eosinophils and/or others should be provided.

6. PLOS authors have the option to publish the peer review history of their article (what does this mean?). If published, this will include your full peer review and any attached files.

Reviewer #1: No

---

## [Author Response · Author response to Decision Letter 0]

21 Mar 2020

All the responds to reviewers and editors were shown in the cover letter and supporting information. Thanks for your comments.

---

## [Decision Letter · Decision Letter 1]

30 Mar 2020

The benefits of higher LMR for early threatened abortion: a retrospective cohort study

PONE-D-19-27160R1

Dear Dr. Gao,

We are pleased to inform you that your manuscript has been judged scientifically suitable for publication and will be formally accepted for publication once it complies with all outstanding technical requirements.

With kind regards,

Frank T. Spradley

Academic Editor

PLOS ONE

Reviewers' comments:

Reviewer's Responses to Questions

**Comments to the Author**

1. If the authors have adequately addressed your comments raised in a previous round of review and you feel that this manuscript is now acceptable for publication, you may indicate that here to bypass the “Comments to the Author” section, enter your conflict of interest statement in the “Confidential to Editor” section, and submit your "Accept" recommendation.

Reviewer #1: All comments have been addressed

2. Is the manuscript technically sound, and do the data support the conclusions?

Reviewer #1: Partly

3. Has the statistical analysis been performed appropriately and rigorously? 

Reviewer #1: Yes

4. Have the authors made all data underlying the findings in their manuscript fully available?

Reviewer #1: Yes

5. Is the manuscript presented in an intelligible fashion and written in standard English?

Reviewer #1: Yes

6. Review Comments to the Author

Reviewer #1: (No Response)

7. PLOS authors have the option to publish the peer review history of their article (what does this mean?). If published, this will include your full peer review and any attached files.

Reviewer #1: No

---

## [Editor Report · Acceptance letter]

8 Apr 2020

PONE-D-19-27160R1 

The benefits of higher LMR for early threatened abortion: a retrospective cohort study 

Dear Dr. Gao:

I am pleased to inform you that your manuscript has been deemed suitable for publication in PLOS ONE. Congratulations! Your manuscript is now with our production department. 

With kind regards,

on behalf of

Dr. Frank T. Spradley 

Academic Editor

PLOS ONE